# *Salmonella* in Captive Reptiles and Their Environment—Can We Tame the Dragon?

**DOI:** 10.3390/microorganisms9051012

**Published:** 2021-05-08

**Authors:** Magdalena Zając, Magdalena Skarżyńska, Anna Lalak, Renata Kwit, Aleksandra Śmiałowska-Węglińska, Paulina Pasim, Krzysztof Szulowski, Dariusz Wasyl

**Affiliations:** 1Department of Microbiology, National Veterinary Research Institute, Al. Partyzantów 57, 24-100 Puławy, Poland; magdalena.skarzynska@piwet.pulawy.pl (M.S.); anna.lalak@piwet.pulawy.pl (A.L.); renata.kwit@piwet.pulawy.pl (R.K.); Aleksandra.Smialowska@piwet.pulawy.pl (A.Ś.-W.); paulina.greszata@piwet.pulawy.pl (P.P.); kszjanow@piwet.pulawy.pl (K.S.); wasyl@piwet.pulawy.pl (D.W.); 2Department of Omics Analyses, National Veterinary Research Institute, Al. Partyzantów 57, 24-100 Puławy, Poland

**Keywords:** *Salmonella*, pet reptile, public health, reptile exhibition, antimicrobial resistance, improved methodology

## Abstract

Reptiles are considered a reservoir of a variety of *Salmonella* (*S*.) serovars. Nevertheless, due to a lack of large-scale research, the importance of *Reptilia* as a *Salmonella* vector still remains not completely recognized. A total of 731 samples collected from reptiles and their environment were tested. The aim of the study was to assess the prevalence of *Salmonella* in exotic reptiles kept in Poland and to confirm *Salmonella* contamination of the environment after reptile exhibitions. The study included *Salmonella* isolation and identification, followed by epidemiological analysis of the antimicrobial resistance of the isolates. Implementation of a pathway additional to the standard *Salmonella* isolation protocol led to a 21% increase in the *Salmonella* serovars detection rate. The study showed a high occurrence of *Salmonella*, being the highest at 92.2% in snakes, followed by lizards (83.7%) and turtles (60.0%). The pathogen was also found in 81.2% of swabs taken from table and floor surfaces after reptile exhibitions and in two out of three egg samples. A total of 918 *Salmonella* strains belonging to 207 serovars and serological variants were obtained. We have noted the serovars considered important with respect to public health, i.e., *S*. Enteritidis, *S*. Typhimurium, and *S*. Kentucky. The study proves that exotic reptiles in Poland are a relevant reservoir of *Salmonella*.

## 1. Introduction

Reptiles are a well-known reservoir of a wide variety of *Salmonella* species, representing numerous subspecies and serovars. These poikilothermic vertebrates can be easily colonized with vertical and horizontal transfer and shed pathogens intermittently [1]. There are many reports of reptile-associated salmonellosis (RAS) in humans, mostly affecting children [2,3,4]. Contact with turtles and tortoises is considered to have a particularly high risk of infection [5,6]. Simultaneously, reptiles are popular pet animals in many countries. In recent decades, this has led to increased importation of reptiles and the creation of reptile breeding farms throughout Poland. Although there are no data from the European Union on the imports, more than 22,700 live reptiles were imported between 2008 and 2015 from non-EU countries, mainly from Africa and Asia (https://www.mos.gov.pl/srodowisko/przyroda/konwencje-miedzynarodowe/kowencja-waszyngtonska-cites/raporty-cites/, accessed on 16 March 2016). The easy availability, low price, and seemingly uncomplicated care and breeding of reptiles have made them popular pets, especially among young customers. Many reptile exhibitions take place in public spaces such as schools, universities, and exhibitions centers, providing the possibility not only to observe and admire the variety of reptile species but also often to touch and hold some of the individuals. Notably, the equipment used for these exhibitions (i.e., tables) is usually used for other daily activities.

The issue of microbiological hazards associated with reptiles kept in houses or zoos is rarely investigated in Poland, but a few reports are available [7,8,9,10]. The aim of this study was to further investigate the occurrence of *Salmonella* in exotic reptiles, including the serovar distribution and antimicrobial resistance of the strains, in the context of an improved detection procedure and public health concerns.

## 2. Materials and Methods

### 2.1. Samples

Over a three-year time period (2011–2013), 731 samples collected from reptiles and their environments were tested. A total of 696 fecal samples were taken from 662 healthy reptiles belonging to 45 species of snakes (*n* = 358; 51.4%), 58 species of lizards (*n* = 276; 39.6%), 24 species of chelonians (*n* = 60; 8.6%), and two crocodiles (*Crocodylus niloticus*). The samples were derived from 10 breeding farms (*n* = 258; 37%), 14 pet shops (*n* = 143; 20.5%), 5 zoological gardens (*n* = 101; 14.5%), 1 reptile shelter (*n* = 98; 14.1%), 3 reptile exhibitions (*n* = 41; 5.9%), and 9 private owners (*n* = 55; 7.9%) (Table 1).

Forty-one of the 696 fecal samples were obtained during a long-term study concerning intermittent *Salmonella* shedding conducted on 10 different reptiles from a reptile shelter and a private owner. Samples were collected up to five times from a single animal in at least two-month intervals.

Thirty-two environmental swabs were taken from four different reptile exhibitions from tables (3 rows, *n* = 24) and floors (boot swabs, *n* = 8). Half of these samples were collected before the start of the reptile exhibitions (after disinfection of the surfaces) and the other half were collected after the exhibitions.

Three pooled samples of unfertilized gecko eggs from 1 breeding farm were investigated. The egg shells and contents were tested separately.

All samples were stored for up to 72 h at 2–8 °C before testing.

### 2.2. Isolation and Identification of Salmonella

*Salmonella* isolation was performed according to the PN-EN ISO 6579:2003/A1:2007 standard. Half of the fecal samples (351/696; 50.4%) were tested using an improved methodology. Besides standard pre-enrichment (Buffered Peptone Water, BWP, Merck, Darmstadt, Germany) followed by selective enrichment (Modified Semisolid Rappaport-Vassiliadis, MSRV, Merck, Germany), it included a simultaneous streak of 10 µL BWP culture on a RAPID’*Salmonella* agar medium (RSA, BIO-RAD, California, CA, USA). After incubation (37 ± 1 °C; 24 ± 3 h), up to three suspected magenta colonies were subcultured on a Xylose Lysine Deoxycholate agar medium (XLD, Oxoid, Hampshire, UK). In the standard approach, the XLD medium was complemented with BxLH (in-house, PIWet, Puławy Poland) [11]. This led to a selection of several colonies with typical *Salmonella* morphology being subcultured on nutrient agar (BioMaxima, Lublin Poland) based on tiny differences in their size, shape, or color intensity. Biochemical confirmation was performed with conventional tests described elsewhere [12,13,14]. In-house media were applied, as well as multiplex PCR [15], which was used to reveal or confirm unclear biochemical confirmation of *Salmonella* species and subspecies. Serotyping was carried out according to the White–Kauffmann–Le Minor scheme [13]. To avoid duplicates, if isolates representing the same serogroup were detected on both MSRV and RSA, those from MSRV were selected for further investigation. *Salmonella* strains were stored (2–8 °C) for further testing and deep-frozen until use.

### 2.3. Antimicrobial Susceptibility Testing

A subset of strains (*n* = 533) was selected for testing antimicrobial resistance based on the serovar and isolation source and location. In the case of the presence of different serovars in the same sample, strains representing each of the serovars were tested. Susceptibility testing was performed using the microbroth dilution method [16] (Sensititre EUVMS plates; TREK Diagnostic Systems, ThermoFisher Scientific, Waltham, MA, USA) for 14 compounds representing eight antimicrobial classes: beta-lactams (ampicillin, cefotaxime, and ceftazidime), quinolones (nalidixic acid), fluoroquinolones (ciprofloxacin), phenicols (chloramphenicol and florfenicol), aminoglycosides (gentamycin, kanamycin, and streptomycin), folate pathway inhibitors (trimethoprim and sulfamethoxazole), tetracyclines (tetracycline), and polymyxins (colistin). Strains were considered microbiologically resistant (non-wild-type, NWT) when the minimum inhibitory concentration (MIC) for each antimicrobial substance was above the epidemiological cut-off value (EUCAST, http://www.eucast.org/mic_distributions_and_qc/, accessed on 27 January 2021) The WhoNet (v.5.6) software was used for MIC data management (https://www.who.int/medicines/areas/rational_use/AMR_WHONET_SOFTWARE/en/, accessed on 27 January 2021).

### 2.4. Statistical Analysis

The KyPlot software (v.5.0) was used to perform χ^2^ independence test calculations.

## 3. Results

### 3.1. Salmonella Occurrence and Serovar Distribution

Overall, *Salmonella* was detected in 85.8% (597/696; 95% CI = 83.2 ÷ 88.4%) of fecal samples, found in 92.2% (89.4 ÷ 95.0%) of snakes, 83.7% (79.3 ÷ 88.1%) of lizards, and 60% (47.6 ÷ 72.4%) of chelonians. Both crocodile samples were negative. Differences between the taxa were statistically significant (*p* ≤ 0.001).

The occurrence of *Salmonella*, depending on the place of sampling, ranged from 74.3% in zoos to 91.8% in the reptile shelter (*p* > 0.05) (Figure 1). All surface swabs taken before the reptile exhibitions were negative for *Salmonella*, but the pathogen was confirmed in 81.2% (13/16) of swabs taken after the exhibitions. Two out of the three egg samples also contained *Salmonella*. Longitudinal testing of selected animals revealed that nine had shed *Salmonella* constantly, and one was negative throughout the whole study period. The number of positive samples and the obtained serovars differed between individuals and ranged from one in the mourning gecko to five in the savannah monitor and ground rattlesnake (Table 2). In two reptiles (mourning gecko and African puff adder), one serovar was isolated throughout the study period.

Environmental samples yielded 27 strains belonging to 21 different serovars (Table 3). The number of strains was different depending on the reptile exhibition, ranging from 2 to 10.

A total of nine strains belonging to six different serovars within two subspecies of *S. enterica* were obtained from the egg samples (Table 4). Different serovars were found in the egg contents and shells. *S*. II 50:b:z_6_ was isolated exclusively from the egg content, whereas *S*. Tennessee was also found in the eggshells. All of the strains found in these samples were also isolated from the feces of reptiles from the same reptile farm.

In 30.8% (*n* = 184) of fecal samples obtained from lizards (*n* = 80), snakes (*n* = 96), and chelonians (*n* = 8), two different *Salmonella* serovars per sample were found. Three serovars per sample were detected in 7.5% (*n* = 44) of samples, and four different serovars were identified in three samples from snakes and a lizard.

From 612 positive samples, 918 *Salmonella* strains belonging to 207 serovars and serological forms were identified. Among the 534 strains obtained from samples tested with the modified isolation method, 113 (21%) were detected on RSA plates and not isolated on MSRV medium from the same sample (Appendix A). Most of those strains belonged to *S*. II 58:a:z_6_ (*n* = 8), *S*. Newport (*n* = 8), *S*. II l,z_13_,z_28_:z_6_ (*n* = 7), and *S*. Fluntern (*n* = 5). Forty-six serovars were represented with single strains. Strains belonging to *S*. II 50:b:z_6_ (*n* = 4), *S*. IIIb 59:k:z (*n* = 2), and *S*. IIIb 59:z_52_:z_53_ (*n* = 2) and some represented by single strains were not isolated on MSRV during the research (Appendix A).

More than 66.4% (*n* = 610) of strains were classified as *S. enterica* subsp. *enterica* (I), followed by subsp. *salamae* (II) (14.6%; *n* = 134), subsp. *diarizonae* (IIIb) (11.2%; *n* = 103), subsp. *arizonae* (IIIa) (4.0%; *n* = 36), and *S. enterica* subsp. *houtenae* (IV) (3.7%; *n* = 34). A single isolate was classified as *S. bongori* 48:z_65_:- (V). The serovars are listed in Table 4. In total, 97% of isolates belonging to *S. enterica* subsp. *arizonae* and 84.0% to *S. enterica* subsp. *diarizonae* were isolated from snakes (*p* ≤ 0.001). *S. enterica* subsp. *salamae* dominated in lizards over snakes (73.2%; *p* ≤ 0.001). The most prevalent serovar was *S.* Oranienburg (*n* = 54), followed by *S*. *enterica* subsp. *salamae* 30:l,z_28_:z_6_ (*n* = 53), *S.* Tennessee (*n* = 46), *S*. Agona (*n* = 43), *S.* Muenchen (*n* = 43), *S*. Fluntern (*n* = 42), and *S.* II 1,40:g,m,t:- (*n* = 29) (Figure 2). *S*. Agona, *S*. Enteritidis, *S*. Muenchen, *S.* Oranienburg, *S.* Newport, and *S.* IIIb 53:z_10_:z_35_ were the most frequent in snakes (respectively: *p* ≤ 0.001, *p* ≤ 0.05, *p* ≤ 0.05, *p* ≤ 0.01, *p* ≤ 0.05, and *p* ≤ 0.01), whereas *S.* Tennessee, *S.* II 30:l,z_28_:z_6,_
*S*. Ago, *S.* Monschaui, and *S.* Fluntern dominated in lizards (respectively: *p* ≤ 0.001, *p* ≤ 0.001, *p* ≤ 0.01, *p* ≤ 0.01, and *p* ≤ 0.01). No serovars were predominant in chelonians.

A single *S. enterica* subsp. *enterica* 47:z_4_,z_23_:- strain was confirmed at Institut Pasteur, Paris, France, as a new *Salmonella* serovar. Six others were auto-agglutinating. Furthermore, 136 serovars (65.7%) were reported for the first time in Poland.

### 3.2. Antimicrobial Resistance

Amongst 533 tested strains, more than 67.2% were susceptible to all tested antimicrobials. Most commonly, the strains were resistant to streptomycin (25.0%), ciprofloxacin (8.1%), and nalidixic acid (8.1%).

Single strains were resistant to tetracycline (*n* = 14; 2.6%), sulfamethoxazole (*n* = 13; 2.4%), ampicillin (*n* = 8; 1.5%), kanamycin (*n* = 5; 0.9%), trimethoprim (*n* = 3; 0.6%), colistin (*n* = 2; 0.4%), chloramphenicol (*n* = 1; 0.2%), and gentamycin (*n* = 1; 0.2%). Resistant isolates belonged mostly to the subspecies *S. enterica* subsp. *enterica*, followed by *S. enterica* subsp. *diarizonae. S. bongori* isolate was susceptible to all tested antimicrobials. Resistance to quinolones in *S. enterica* subsp. *enterica* (12.1%, *p* ≤ 0.001) appeared significantly higher compared to that in other *Salmonella* subspecies. It was also identified as being the most prevalent in breeding farms (16.4%, *p* ≤ 0.05) and in samples taken from snakes (13.2%, *p* ≤ 0.001). Of the public health-relevant serovars deriving from different reptiles, isolates of *S*. Kentucky (*n* = 3) were classified as multi-drug resistant (MDR). Over 90% of *S.* Agona and 60% of *S*. Adelaide isolates were resistant to ciprofloxacin and nalidixic acid, whereas *S*. Typhimurium was susceptible to all tested antimicrobials.

## 4. Discussion

The popularity of captive reptiles as pets is continuously increasing. Therefore, recognition of the hazards associated with pet reptiles is becoming critical to avoid their negative consequences for human health. Our study, covering a broad collection of samples, confirmed that pet reptiles and their environments constitute a considerable reservoir of *Salmonella*. Moreover, a number of *Salmonella* strains noted in this study belonged to serovars of public health concern, i.e., Enteritidis, Typhimurium, Infantis, Hadar, Newport, Oranienburg, and Muenchen. The occurrence of those serovars has also been reported by previous studies on reptiles [4,17,18,19]. It is believed that in some cases, *Salmonella* may have been delivered with poultry meat [7]. However, it should be pointed out that although these were identified as “public health risks”, all serovars of *Salmonella* have the potential to infect humans and result in salmonellosis. There were no confirmed cases of RAS in Poland, but one study suggests an epidemiological relation between *S*. Lindern isolates found in infants and tortoises [20]. In particular, regarding RAS, testing of the reptile exhibition environment samples seems to be crucial. Essentially, the current study has shown that at least two positive samples were obtained after each reptile exhibition. Many *Salmonella* isolates, including *S*. Kentucky and *S.* Enteritidis, were detected both in samples taken from the tables and floor. The use of general purpose school or hall equipment during exhibitions can have adverse consequences for humans. Many authors pay particular attention to the risk posed by *Salmonella* in educational centers where children are in direct contact with reptiles and in places where both humans and animals dwell [21,22,23,24]. Some studies also indicate that in some cases, this pathogen can be less common in reptiles than in the habitat of the reptiles [25]. The high survival rate of *Salmonella* in the environment allows the pathogen to survive in the terrarium long after the infected animals have been removed [26,27]. It follows that direct contact with a reptile is not necessary for the transmission of *Salmonella*.

Similar to other studies, a high prevalence of *Salmonella* was found, particularly in snakes and lizards [28]. According to Kepel et al., representatives of these reptile groups, especially Boinae, Iguanidae, and Chamaelonidae, are the most popular among reptiles available in the country [29]. Isolates belonging to *S. enterica* subsp. *arizonae* and *S. enterica* subsp. *diarizonae* seemed to be snake-related, which has been proven by others [30,31]. The obtained results indicate that *Salmonella* may be present in the majority of reptile-keeping households. Some lizards, such as leopard geckos, are considered easy to breed and are often recommended to people interested in reptiles. Additionally, amongst many reptile farmers, the knowledge about *Salmonella* carriage in reptiles and its possible consequences often remains at a basic level. Our study has shown no differences in the occurrence of this pathogen depending on the place of reptile origin. It is also congruent with others indicating a considerably high occurrence of *Salmonella* among reptiles kept in captivity [17,18,30,32]. A high-density reptile population promotes the transfer of the pathogen between animals. Moreover, using undisinfected equipment and feeding with rodents and one-day-old chicks are additional factors leading to the spread of *Salmonella* [7,33,34,35].

There is little information about the mechanism of vertical transmission of *Salmonella* in reptiles [1,36]. In this study, all *Salmonella* serovars isolated from both the egg contents and the eggshells were also found in the leopard gecko individuals from which the eggs were derived. The presence of this bacterium in egg content might prove the role of vertical transmission in the spread of *Salmonella* in reptile populations, but this should be investigated in future studies. In the case of some reptiles that take care of eggs and young cubs, such as crocodiles or royal cobras, horizontal transmission should also be taken into account. It makes this research area even more interesting and worth exploring.

The rate of *Salmonella* detection can differ significantly by the animal specimen, habitat, type of sample, and, finally, the methodology used [18,25,37]. The additional step in the standardized methodology applied in this study led to an approximately 21% increase in the *Salmonella* serovar detection rate and allowed to identify multiple *Salmonella* serovars in 38% of samples. A possible explanation is various selectivity of the RSA and MRSV media in relation to different serovars [38]. Different numbers of colony-forming units and competition between multiple *Salmonella* strains present in the same mixture may influence the detection of only selected serovars [38]. Confirmation of more than one suspected colony from selective media, especially with visible differences in morphology, can significantly improve the assessment of the real serovar prevalence in reptile samples. The results suggest that if a sample is contaminated by more than one *Salmonella* serovar, some of them might be missed in the standard approach. Therefore, dedicated protocols and special attention for *Salmonella* detection in reptiles should be recommended.

In contrast to the results obtained by Goupil et al., indicating periodic shedding of *Salmonella* in snake feces, our research showed that the majority of the animals tested excreted the pathogen constantly [33]. This may be associated with the mostly bad health condition of the reptiles and the high level of stress occurring in animals submitted to the reptile shelter. Simultaneously, in most cases, one serovar was dominant and was found several times during the research interval, which has also been observed in other studies [33]. The obtained results suggest that *Salmonella* serovar diversity can be very high in an individual, but only multiple testing of the animal presents the possibility to prove this.

Overall, the antimicrobial resistance of the *Salmonella* found in reptiles remained low compared to food-producing animals [39]. Single multidrug-resistant isolates were found in different reptile species, with *S*. Kentucky being found most often [7]. A possible reason for the increasing antimicrobial resistance in reptile isolates may be the overuse of such drugs during treatment, or feeding with rodents or meat contaminated with MDR *Salmonella* isolates [40]. Therefore, carnivore reptiles should be taken into account as a possible vector of infection with multidrug-resistant *Salmonella.*

## 5. Conclusions

The study demonstrates the important role of reptiles as a reservoir for *Salmonella*, representing a variety of serovars, susceptible or multidrug resistant, being a potential hazard to humans. The obtained results justify the need for the education of reptile owners and monitoring of the occurrence of *Salmonella* in their pets. Every potential owner should be advised on the risk of *Salmonella* infection. If an event of diarrheal or bacteremic disease occurs in the family or other persons in contact with the animal, the doctor should be informed about this potential source of infection. This also applies to reporting attendance at reptile exhibitions. On the other hand, attention should be paid to proper decontamination of the environment and equipment following such events. Introduction of dedicated disinfection procedures would limit the risk of not only RAS but also other zoonotic bacteria often found in reptiles. Considering the diversity of *Salmonella* as well as the impressive biodiversity, behavior, and habitat of reptiles, they will always be an intriguing object of fundamental studies.

## Figures and Tables

**Figure 1 microorganisms-09-01012-f001:**
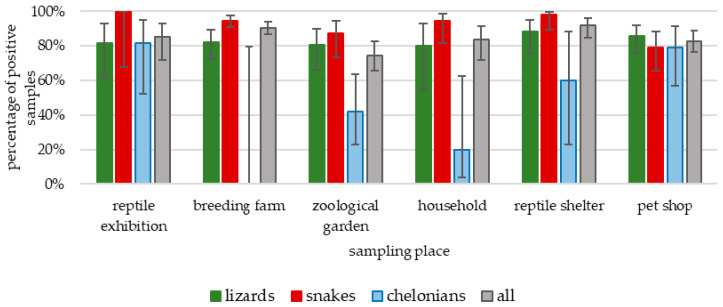
Prevalence of *Salmonella* by animal taxa and sampling place.

**Figure 2 microorganisms-09-01012-f002:**
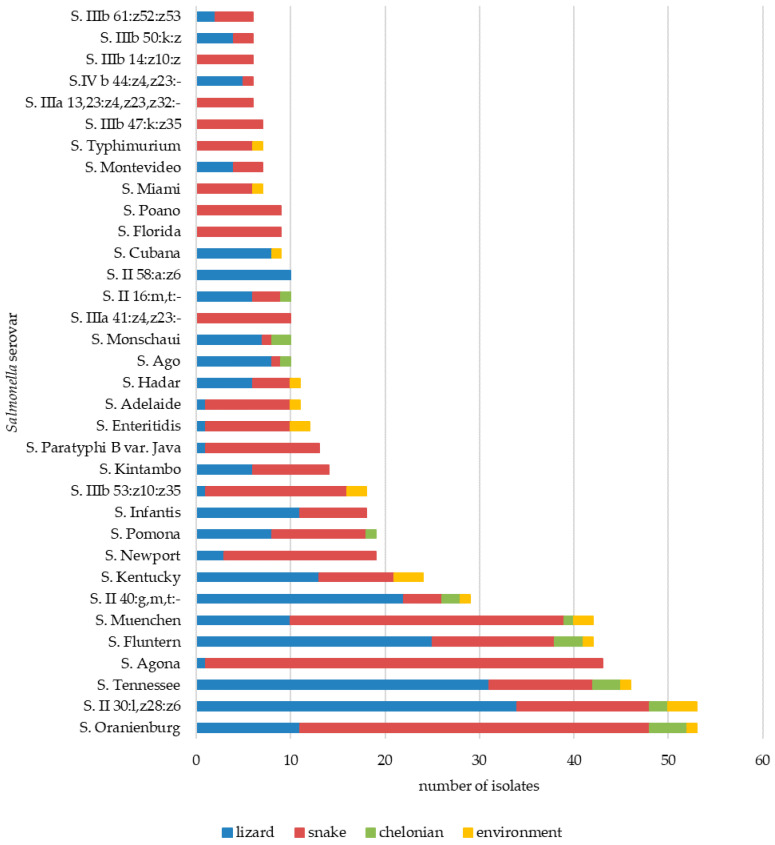
Distribution of the most prevalent *Salmonella* serovars found in animal taxa and their environments.

**Table 1 microorganisms-09-01012-t001:** Number of tested samples regarding the sampling place and reptile taxa.

Sampling Place	Reptile Group				Environment	Eggs	Total
	Chelonian	Crocodile	Lizard	Snake			
Breeding farm	1		79	178		3	258
Pet shop	19		76	48			143
Private household	5		15	35			55
Reptile exhibition	22		8	11	32		41
Reptile shelter	5		43	50			98
Zoological garden	19	2	41	39			101
Total	60	2	276	358			
	696				32	3	731

**Table 2 microorganisms-09-01012-t002:** *Salmonella* serovars found in fecal samples of reptiles tested periodically.

Sampling Place	Reptile Species	*Salmonella* Serovar	Sampling No.				
			1	2	3	4	5
Reptile shelter	Mexican kingsnake (*Lampropeltis mexicana*)	Fluntern	x	x			
		Tennessee		x		x	
		II 30:l,z_28_:z_6_			x		
		IIIb 14:z_10_:z			x		x
Reptile shelter	Saharan horned viper (*Cerastes cerastes*)	IIIb 57:k:e,n,x,z_15_	x				
		IIIb 53:z_10_:z_35_		x	x	x	x
		Fluntern				x	
		II 30:l,z_28_:z_6_				x	
Reptile shelter	Ground rattlesnake (*Sistrurus miliarius*)	Agona	x				
		II 30:l,z_28_:z_6_		x	x	x	x
		Mundonobo		x			x
		IIIb 59:k:z		x			
		IIIb 59:z_52_:z_53_					x
Reptile shelter	Horned viper (*Vipera ammodytes*)	IIIb 57:l,v:z_35_	x				
		II 30:l,z_28_:z_6_		x	x		
		IIIb 59:k:z			x		
Reptile shelter	Green iguana (*Iguana iguana*)	II 30:l,z_28_:z_6_	x	x		x	x
		Tennessee			x	x	x
Reptile shelter	Savannah monitor (*Varanus exanthematicus*)	Jangwani	x				
		Cubana		x	x		
		Overschie		x		x	
		IIIb 50:z:z_52_		x			
		Tennessee				x	
Reptile shelter	Mourning gecko (*Lepidodactylus lugubris*)	Infantis	x	x	x	x	x
Reptile shelter	Indian python (*Python molurus*)	IV 42:z_36_:-	x				
		Fluntern		x	x	x	
		Infantis			x		
		Redlands				x	
Private household	Russian tortoise (*Testudo horsfieldii*)	-	-	-	-		
Reptile shelter	African puff adder (*Bitis arietans*)	Muenchen	x	x	x	x	x
		IIIb 57:k:e,n,x,z_15_		x	x		
		IIIb 50:r:z			x		

**Table 3 microorganisms-09-01012-t003:** *Salmonella* serovars isolated from swabs taken after the reptile exhibitions.

Sampling Site	Exhibition No. 1	Exhibition No. 2	Exhibition No. 3	Exhibition No. 4
Tables—row no. 1	IIIa 41:z_4_,z_23_:-	IIIb 53:z_10_:z_35_ Tennessee Adelaide	-	-
Tables—row no. 2	II 30:l,z_28_:z_6_,	Enteritidis Typhimurium IV 48:z_4_,z_32_:-	Kentucky	Kentucky Fresno Hadar II 30:l,z_28_:z_6_ IIIb 53:z_10_:z_35_
Tables—row no. 3	Tsevie, Apeyeme	Oranienburg II 1,40:g,m,t	-	Kentucky
Floor	Fluntern, Ituri, IIIb 48:z_52_:z	Enteritidis V 48:z_4_,z_32_:-	II 41:g,t:-	Miami Muenchen

**Table 4 microorganisms-09-01012-t004:** *Salmonella* serovars found in fecal samples and eggs.

	Source of Isolation	Fecal Samples	Unhatched Eggs
*Salmonella enterica* subsp.	*enterica* (610)	Abony (1), Adelaide (10), Ago (10), Agona (43), Alachua (3), Anatum (1), Apapa (4), Aqua (6), Baildon (1), Bardo (1), Bareilly (1), Benin (4), Bispebjerg (1), Blijdorp (2), Blukwa (1), Bolombo (2), Braenderup (3), Brandenburg (1), Carrau (5), Chicago (1), Choleraesuis var. Decatur (1), Cotham (3), Cubana (8), Derby (2), Durban (1), Eastbourne (2), Ekpoui (3), Enteritidis (10), Florida (9), Fluntern (39), Fomeco (1), Fresno (3), Gaminara (1), Gatuni (5), Glostrup (3), Hadar (10), Hofit (1), Ilala (1), Infantis (18), Inverness (2), Itami (1), Jangwani (3), Jodhpur (2), Johannesburg (2), Kentucky (20), Kintambo (14), Kisarawe (1), Koketime (1), Labadi (1), Larochelle (1), Lattenkamp (6), Lisboa (1), Lome (1), Madelia (1), Manhattan (2), Miami (6), Minnesota (2), Monschaui (10), Montevideo (7), Mountpleasant (1), Muenchen (41), Muenster (2), Mundonobo (5), Naware (1), Newport (19), Nima (6), Oranienburg (53), Oritamerin (1), Orlando (1), Oslo (2), Othmarschen (1), Overschie (2), Panama (2), Paratyphi B v. Java (13), Patience (1), Poano (9), Pomona (19), Poona (3), Reading (1), Redlands (1), Rosslyn (2), Saintpaul (1), Sandiego (1), Senftenberg (2), Singapore (2), Tanzania (1), Teddington (3), Telelkebir (3), Tennessee (43), Tonev (1), Toucra (1), Treforest (1), Typhimurium (6), Urbana (3), Uzaramo (2), Virginia (1), 35:-:- (1), 4,5:b:- (6), 4:eh:- (6), 45:b:- (5), 47:z_4_,z_23_:- (1), 6,8:-:- (1), *Salmonella* sp. (rough) (3)	Tennessee (2), Fluntern (2), Fresno (1), Kentucky (1)
	*salamae* (134)	9:a:1,5 (1), 9:z_29_:1,5 (1), 9,46:z:- (1), 9,46:z_10_:- (1), 11:z:e,n,x (1),16:m,t:- (10), 16:t:- (3), 17:g,t:- (2), 21:g,t:- (3), 21:m,t:- (1), 21:z_10_:- (2), 21:z_10_:z_6_ (2), 30:l,z_28_:z_6_ (50), 40:g,m,t:- (27), 40:z_10_:e,n,x (1), 43:g,m,t:- (1), 47:a:1,5 (1), 47:b:e,n,x,z_15_ (1), 50:b:z_6_ (3), 58:a:z_6_ (10), 58:l,z_13_,z_28_:- (1), 58:l,z_13_,z_28_:z_6_ (3), 58:z_39_:e,n,x,z_15_ (1)	40:g,m,t:- (2), 50:b:z_6_ (1),
	*arizonae* (36)	13,23:z_4_,z_23_,z_32_:- (6), 13,23:z_4_,z_32_:- (1), 40:z_4_,z_23_,z_32_:- (1), 41:z_4_,z_23_:- (10), 42:z_4_,z_24_:- (1), 44:z_4_,z_23_,z_32_:- (4), 44:z_4_,z_23_:- (1), 44:z_4_,z_24_:- (1), 44:z_4_,z_32_:- (2), 48:g,z_51_:- (2), 48:z_4_,z_24_:- (2), 51:z_4_,z_23_:- (1), 54:z_4_,z_23_,z_32_:- (1), 56:z_4_,z_23_,z_32_:- (1), *Salmonella* sp. (rough) (1)	
	*diarizonae* (103)	6,14:z_10_:z (1), 11:l,v:z (1), 14:z_10_:z (6), 18:l,v:z (1), 35:i:z_35_ (1), 35:k:z_53_ (1), 35:l,v:z_35_ (1), 38:-:z (1), 38:k:1,5,7 (3), 38:r:1,5,7 (1), 38:r:z (2), 42:l,v:1,5 (1), 43:r:z_53_ (1), 47:k:z_35_ (7), 47:l,v:z (1), 47:r:z_53_ (2), 47:z_10_:z_35_ (1), 48:-:- (1), 48:i:z (4), 48:k:z_53_ (3), 48:r:z (1), 48:z_4_,z_24_:- (1), 48:z_52_:z (1), 50-:- (1), 50:i:1,5,7 (1), 50:k:z (7), 50:r:- (2), 50:r:z (1), 50:z:z_52_ (2), 50:z_52_:z_53_ (2), 51:k:z_35_ (1), 53:z_10_:z_35_ (16), 57:k:e,n,x,z_15_ (4), 57:l,v:z_35_ (1), 58:r:z_53_ (1), 58:z_52_:z_35_ (1), 59:k:z (2), 59:z_52_:z_53_ (4), 61:i:z (1), 61:l,v:1,5 (1), 61:z_52_:z_53_ (6), 65:z_10_:e,n,x,z_15_ (1), 65:z_52_:z (1), *Salmonella* sp. (rough) (1)	
	*houtenae* (34)	11:z_4_,z_23_:- (2), 16:z_4_,z_32_:- (2), 38:z_4_,z_23_:- (4), 40:z_4_,z_24_,- (1), 41:z_4_,z_23_:- (1), 42:z_36_:- (3), 43:z_4_,z_23_:- (2), 44:z_4_,z_23_:- (6), 44:z_4_,z_24_:- (1), 45:g,z_51_:- (3), 48:g,z_51_:- (2), 50:g,z_51_:- (1), 51:z_4_,z_23_:- (1), 53:g,z_51_:- (2), *Salmonella* sp. (rough) (1)	

## Data Availability

The data presented in this study are included in the manuscript and Appendix A.

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
