# Peer review of "Salmonella in Captive Reptiles and Their Environment—Can We Tame the Dragon?"

_microorganisms, 2021, doi:10.3390/microorganisms9051012_

Round 1
Reviewer 1 Report
In the manuscript ID microorganisms-1184542 by ZajÄ…c and colleagues, the authors have highlighted the incidence of Salmonella spp in reptile pets in Poland, performing the pathogen isolation, identification and typing from a wide variety of sources, mostly (696/731) faecal samples from the animals. The authors managed to detect Salmonella spp in most samples and to evidence the occurrence of serovars relevant for human health, thus proving the importance of reptiles as vectors for salmonellosis. Remarkably, they demonstrated the danger related to the environment contamination, and the consequent spread of pathogenic clones, due to reptile exhibitions and the possibility of vertical transmission in reptile eggs.
Although not directly connected to foodborne pathogenesis, the presented topic is consistent with the focus of the proposed special issue, mainly the dissemination of pathogenic clones, the risk factors related to zoonosis and salmonellosis control. The paper is clear, the results are well discussed, supported by a consistent literature and statistical analysis.
As minor concerns:
-The Salmonella vertical transmission through reptile eggs is interesting, but, as acknowledged by the authors, the sample number (3) is too limited to drive conclusions. The authors should consider reporting these data as suggestions (i.e correct “might prove the role” in line 224) and, possibly, to investigate these finding more in detail in future studies;
-Please provide better quality tables (1 and 2), using the same font, and figures, reporting the axis titles.
After these minor revisions, the paper can be considered suitable for publication in “Microorganisms”.
MINOR COMMENTS
Line 11, please correct “variety of Salmonella (S) serovars”;
Line 15, please correct “confirmation of Salmonella contamination” and “after reptile exhibitions”;
Line 33, please correct “In particular, contact with turtles and tortoises is considered a high risk”;
Line 36, please delete the space before “above”;
Lines 44, 45, please correct “is usually used for other”;
Lines 75-79, please define the acronyms “BWP, MSRV, XLD and BXLH”;
Line 120, please correct “one serovar”;
Line 138, please correct “the modified isolation method”;
Line 185, please correct “The occurrence of those serovars was also reported”;
Line 189, please delete the space before “suggests”;
Line 192, please correct “the current study”;
Lines 217-219, please delete the sentence “Isolates belonging to…proved by others”, as it is already reported above;
Lines 238 and 247, please correct “The results suggest”;
Line 247, please correct “which was also observed”;
Line 265, please correct “to proper equipment”.
Author Response
Response to Reviewer 1
As minor concerns:
-The Salmonella vertical transmission through reptile eggs is interesting, but, as acknowledged by the authors, the sample number (3) is too limited to drive conclusions. The authors should consider reporting these data as suggestions (i.e correct “might prove the role” in line 224) and, possibly, to investigate these finding more in detail in future studies;
The sentence below was removed:
”Despite the small number of tested samples, the pathogen was not found in only one of them”
The sentences were changed as follows:
The presence of this bacterium in egg content might prove the role of vertical transmission in the spread of Salmonella in reptile populations but this should be investigated in the future studies. In the case of some reptiles that take care of eggs and young cubs, such as crocodiles or royal cobras, horizontal transmission should also be taken into account. It makes this research area even more interesting and worth to explore.”
-Please provide better quality tables (1 and 2), using the same font, and figures, reporting the axis titles.
All suggested changes were amended.
After these minor revisions, the paper can be considered suitable for publication in “Microorganisms”.
MINOR COMMENTS
Line 11, please correct “variety of Salmonella (S) serovars”;
corrected
Line 15, please correct “confirmation of Salmonella contamination” and “after reptile exhibitions”;
corrected
Line 33, please correct “In particular, contact with turtles and tortoises is considered a high risk”;
corrected
Line 36, please delete the space before “above”;
corrected
Lines 44, 45, please correct “is usually used for other”;
corrected
Lines 75-79, please define the acronyms “BWP, MSRV, XLD and BXLH”;
The full names for abbreviations were added:
“Besides standard pre-enrichment (Buffered Peptone Water, BWP, Merck) followed by selective enrichment (Modified Semisolid Rappaport-Vassiliadis, MSRV, Merck, Ger-many), it included a simultaneous streak of 10µl BWP culture on a RAPID Salmonella agar medium (RSA, BIO-RAD, USA). After incubation (37 ± 1°C; 24 ± 3h), up to three suspected magenta colonies were subcultured on a Xylose Lysine Deoxycholate agar (XLD, Oxoid, Great Britain).
The only exception is BxLH. It is the in-house produced medium patented in in 1993 under the BxLH name (patent No. 158862) and used in our lab ever since. The first research paper mentioning the medium was published at Comparative Immunology, Microbiology and Infectious Diseases 1995, 18(4): 272-237 - the reference was added.
Line 120, please correct “one serovar”;
corrected
Line 138, please correct “the modified isolation method”;
corrected
Line 185, please correct “The occurrence of those serovars was also reported”;
corrected
Line 189, please delete the space before “suggests”;
corrected
Line 192, please correct “the current study”;
corrected
Lines 217-219, please delete the sentence “Isolates belonging to…proved by others”, as it is already reported above;
The repeated sentence was removed
Lines 238 and 247, please correct “The results suggest”;
corrected
Line 247, please correct “which was also observed”;
corrected
Line 265, please correct “to proper equipment”.
corrected
The manuscript was corrected by English native speaker
Reviewer 2 Report
General comments
In my opinion this article needs a lot of language editing, preferable by native speaking
Section 4 (Discussion) needs a lot of work; there are many syntax errors, many sentences with confusing meaning
Lines 75-79
Please provide whole names for abbreviations, since this is the first time they appear
(BWP, MSRV, RSA, XLD, BxLH)
(eg MSRV=modified semisolid Rappaport-Vassiliadis)
Lines 74-82
Please provide references for the improved methodology
Lines 93-97
Please provide refs for the microbroth dilution method
Please provide more information on … 14 compounds representing 8 antimiclobial …
MIC, please state (Minimal Inhibitory Concentration)
Lines 163-177
A table or figure summarizing this info could be very helpful
Author Response
Response to Reviewer 2
Thank you for your time and input in improvement of this manuscript
General comments
In my opinion this article needs a lot of language editing, preferable by native speaking
Section 4 (Discussion) needs a lot of work; there are many syntax errors, many sentences with confusing meaning
The manuscript was submitted to language corrections
Lines 75-79
Please provide whole names for abbreviations, since this is the first time they appear
(BWP, MSRV, RSA, XLD, BxLH)
(eg MSRV=modified semisolid Rappaport-Vassiliadis)
The full names for abbreviations were added:
“Besides standard pre-enrichment (Buffered Peptone Water, BWP, Merck) followed by selective enrichment (Modified Semisolid Rappaport-Vassiliadis, MSRV, Merck, Ger-many), it included a simultaneous streak of 10µl BWP culture on a RAPID Salmonella agar medium (RSA, BIO-RAD, USA). After incubation (37 ± 1°C; 24 ± 3h), up to three suspected magenta colonies were subcultured on a Xylose Lysine Deoxycholate agar (XLD, Oxoid, Great Britain).
The only exception is BxLH. It is the in-house produced medium patented in in 1993 under the BxLH name (patent No. 158862) and used in our lab ever since. The first research paper mentioning the medium was published at Comparative Immunology, Microbiology and Infectious Diseases 1995, 18(4): 272-237 - the reference was added.
Lines 74-82
Please provide references for the improved methodology
There is no reference, this method was applied for the first time in this study
Lines 93-97
Please provide refs for the microbroth dilution method
The missing reference was added:
International Organization for Standardization ISO 20776-1:2006 Susceptibility testing of infectious agents and evaluation of performance of antimicrobial susceptibility test devices — Part 1: Broth micro-dilution reference method for testing the in vitro activity of antimicrobial agents against rapidly growing aerobic bacteria involved in infectious diseases
Please provide more information on … 14 compounds representing 8 antimiclobial …
MIC, please state (Minimal Inhibitory Concentration)
Information was added:
for 14 compounds representing eight antimicrobial classes: beta-lactams (ampicillin, cefotaxime, ceftazidime), quinolones (nalidixic acid), fluoroquinolones (ciprofloxacin), phenicols (chlo-ramphenicol, florfenicol), aminoglycosides (gentamycin, kanamycin, streptomycin), folate-path inhibitors (trimethoprim, sulfamethoxazole), tetracyclines (tetracycline), and polymyxins (colistin). Strains were considered microbiologically resistant (non-wild type, NWT) when the Minimum Inhibitory Concentration (MIC) for each antimicrobial substance was above the epidemiological cut-off value (EUCAST, http://www.eucast.org/mic_distributions_and_qc/).
Lines 163-177
A table or figure summarizing this info could be very helpful
We did not summarize those data in a tables/figures to avoid too many graphics in a manuscript
Presenting those data in graphical form require to include 3 figures/tables. First regarding to resistance in overall, second regarding to Salmonella subspecies, and third regarding to sampling place and animal taxa. In this paragraph we extracted the most important data which contribute to expand the knowledge.
Round 2
Reviewer 1 Report
In the revised version of the manuscript ID microorganisms-1184542 by ZajÄ…c and colleagues, the authors have successfully addressed all the raised concerns and performed all the corrections suggested, thus improving the quality of the paper, which can now be published in “Microorganisms”.
Reviewer 2 Report
revision accepted